# Combining genomic and epidemiological data to compare the transmissibility of SARS-CoV-2 variants Alpha and Iota

Mary E. Petrone [1,15✉], Jessica E. Rothman[1,15], Mallery I. Breban [1], Isabel M. Ott[1], Alexis Russell [2], Erica Lasek-Nesselquist[2,3], Hamada Badr [4], Kevin Kelly[5], Greg Omerza[5], Nicholas Renzette[5], Anne E. Watkins [1], Chaney C. Kalinich[1], Tara Alpert[1], Anderson F. Brito[1], Rebecca Earnest[1], Irina R. Tikhonova[6], Christopher Castaldi[6], John P. Kelly[2], Matthew Shudt[2], Jonathan Plitnick[2,3], Erasmus Schneider[2,3], Steven Murphy[7], Caleb Neal[7], Eva Laszlo[7], Ahmad Altajar[7], Claire Pearson[8], Anthony Muyombwe[8], Randy Downing[8], Jafar Razeq[8], Linda Niccolai[1], Madeline S. Wilson[9], Margaret L. Anderson[1], Jianhui Wang[10], Chen Liu[10], Pei Hui[10], Shrikant Mane[6], Bradford P. Taylor[11], William P. Hanage [11], Marie L. Landry[12], David R. Peaper[12], Kaya Bilguvar[6,13], Joseph R. Fauver [1], Chantal B. F. Vogels [1], Lauren M. Gardner[4], Virginia E. Pitzer [1], Kirsten St. George [2,3], Mark D. Adams [5] & Nathan D. Grubaugh [1,14✉]

SARS-CoV-2 variants shaped the second year of the COVID-19 pandemic and the discourse around effective control measures. Evaluating the threat posed by a new variant is essential for adapting response efforts when community transmission is detected. In this study, we compare the dynamics of two variants, Alpha and Iota, by integrating genomic surveillance data to estimate the effective reproduction number ($R_t$) of the variants. We use Connecticut, United States, in which Alpha and Iota co-circulated in 2021. We find that the $R_t$ of these variants were up to 50% larger than that of other variants. We then use phylogeography to show that while both variants were introduced into Connecticut at comparable frequencies, clades that resulted from introductions of Alpha were larger than those resulting from Iota introductions. By monitoring the dynamics of individual variants throughout our study period, we demonstrate the importance of routine surveillance in the response to COVID-19.

[1] Department of Epidemiology of Microbial Diseases, Yale School of Public Health, New Haven, CT 06510, USA. [2] Wadsworth Center, New York State Department of Health, Albany, NY 12208, USA. [3] Department of Biomedical Sciences, University at Albany, SUNY, Albany, NY 12222, USA. [4] Department of Civil and Systems Engineering, Johns Hopkins University, Baltimore 21218 MD, USA. [5] The Jackson Laboratory for Genomic Medicine, Farmington, CT 06032, USA. [6] Yale Center for Genome Analysis, Yale University, New Haven, CT 06510, USA. [7] Murphy Medical Associates, Greenwich, CT 06830, USA. [8] Connecticut State Department of Public Health, Rocky Hill, CT 06067, USA. [9] Yale Health Center, Yale University, New Haven, CT 06510, USA. [10] Department of Pathology, Yale University School of Medicine, New Haven, CT 06510, USA. [11] Center for Communicable Disease Dynamics, Department of Epidemiology, Harvard T. H. Chan School of Public Health, Boston, MA 02115, USA. [12] Departments of Laboratory Medicine and Medicine, Yale University School of Medicine, New Haven, CT 06510, USA. [13] Department of Genetics, Yale University School of Medicine, New Haven, CT 06510, USA. [14] Department of Ecology and Evolutionary Biology, Yale University, New Haven, CT 06510, USA. [15] These authors contributed equally: Mary E. Petrone, Jessica E. Rothman. ✉email: mary.petrone@yale.edu; nathan.grubaugh@yale.edu

The emergence of novel SARS-CoV-2 variants has shaped the second year of the COVID-19 pandemic[1–3] and illustrated the role of genomic epidemiology in facilitating an appropriate, effective, and timely public health response[4]. In particular, genomic epidemiology can determine the source and frequency of new variant introductions into a community, thus indicating where additional surveillance is needed. However, this assessment requires the prior establishment of a robust genomic surveillance system. Once community transmission is documented, the efficacy of control methods should be re-evaluated by assessing the public health risk posed by the variant in comparison to other variants in circulation. This second objective is particularly challenging because factors other than the virus genotype influence its transmission and spread[5–7]. Specifically, competition between virus lineages, human behavior, and local levels of immunity could impact the relative success of a new variant compared to its predecessors. Therefore, to measure relative differences in intrinsic viral properties such as immune evasion and replication rates, we should compare lineages that have emerged concurrently in the same human and virus population as the variant under scrutiny. Instances in which these criteria are met are both rare and exceptionally informative.

At the beginning of 2021 two variants of public health concern synchronously emerged in Connecticut, a United States (US) state with high rates of SARS-CoV-2 genomic surveillance. SARS-CoV-2 variant Iota (lineage B.1.526) was detected in New York in December 2020[8]. Shortly thereafter, cases of Alpha (lineage B.1.1.7), the variant first characterized in the United Kingdom, were identified in the northeastern US. Due to evidence collected in the United Kingdom that this variant was more transmissible than other lineages, Alpha was expected to become dominant in the US by March[9–11]. Instead, Iota co-circulated in New York with Alpha and may have slowed the decline of COVID-19 incidence in New York City[12]. Both variants were initially detected in Connecticut within the first two weeks of January 2021, likely introduced by infected travelers, and continued to co-circulate in the state for months[10].

In this study, we assess the relative dynamics of Alpha and Iota by combining epidemiological and genomic data collected in Connecticut between January and May 2021. We first measure the relative transmissibility, which we herein define as, collectively, the intrinsic viral properties that give rise to secondary infections, of these variants by modeling their growth rates and time-varying effective reproduction numbers following their emergence. Both metrics indicate that Alpha and Iota were up to 50% more transmissible than other lineages that circulated in the same population. Interestingly, these findings are consistent with the relationship we observed in New York City where Iota was established before Alpha. We next estimate the timing, number, and clade size following sustained introductions of each variant into Connecticut to determine whether the apparent fitness advantage we observed for Alpha could be attributed to a higher rate of introductions over our study period rather than higher fitness. We use discrete phylogeography to infer the source and number of introductions for each variant and find that both were introduced at comparable rates, but the size of clades precipitated by introductions of Alpha were on average larger than those formed from introductions of Iota. The concordance of our epidemiological and phylodynamic results indicate that Alpha had a fitness advantage over Iota when potentially confounding factors were controlled.

## Results

### Rapid rise in Alpha and Iota prevalence in Connecticut and New York City.

The rapid spread of SARS-CoV-2 variants Alpha in the United Kingdom[13] and Iota in New York City[12,14] suggested that these variants have an advantage over other SARS-CoV-2 lineages. Both variants are defined by key amino acid substitutions in the spike protein that may contribute to this advantage. We therefore hypothesized that Alpha and Iota would become dominant in Connecticut soon after they emerged. To test this hypothesis, we measured the daily frequencies and growth rates of Alpha and Iota in Connecticut and compared these patterns to those observed in New York City (Fig. 1). Our analysis revealed that Alpha and Iota displaced nearly all other lineages circulating in both regions within three months of emergence. Moreover, the frequency of Alpha grew at a faster rate than Iota.

Unlike the situation in New York City, which may be the origin of Iota, Alpha and Iota emerged concurrently in Connecticut through infected travelers. Connecticut is a state in the northeast US, bordered by Rhode Island, Massachusetts, and New York (Fig. 1a, map). These states experienced synchronous waves of COVID-19 incidence throughout the pandemic (Fig. 1a, graphs). We first detected Alpha in Connecticut on January 6, 2021 (sample collection date) in New Haven County[10], and we detected the first Iota genome soon after on January 14, 2021. Due to the concurrent introductions of these variants into New Haven County, we assumed that any observed differences in fitness between the two variants could not be attributed to a pure founder effect.

The nomenclature used to define the Iota clade has changed throughout the course of the pandemic. For a brief period, the Iota lineage B.1.526 was partially split into two sublineages, B.1.526.1 and B.1.526.2, to account for occurrence of three key amino acid substitutions in the *spike* gene: L452R, S477N, and E484K (Supplementary Fig. 1a)[12,14–17]. However, the sublineages had poor phylogenetic resolution and were difficult to consistently classify by pangolin[18]. As a result, the sublineage designations were removed and all sequences within this clade were reclassified as "B.1.526". While B.1.526 sequences with different combinations of L452R, S477N, and E484K substitutions may have different phenotypes, we previously did not find significant differences among these genotypes at reducing neutralizing antibody titers following vaccination[19]. Furthermore, we found that the B.1.526 genotype frequencies were relatively stable over time in Connecticut (Supplementary Fig. 1b), suggesting that they did not have significant differences in transmissibility. We therefore elected to analyze the dynamics of the B.1.526 genotypes collectively, which we hereafter refer to as Iota.

To compare the relative growth rates of Alpha and Iota over time, we collected and sequenced 2,951 whole SARS-CoV-2 genomes from Connecticut between November 30, 2020 and May 9, 2021 using an unbiased sampling approach. Specifically, we excluded genomes that were targeted for sequencing because of *spike*-gene target failure or any other anomaly. We assigned PANGO lineages to each genome[20] and created a general lineage classification with three categories: 'Alpha' (lineage B.1.1.7), 'Iota' (lineage B.1.256), or 'other'. The 'other' lineages primarily include lineages B.1.2, B.1.517, B.1.575, and B.1.243, but they also include low frequencies of several 'Variants Being Monitored' (VBMs) and Variants of Interest (VOIs; Supplementary Table 1). We calculated a rolling 7-day average for each general lineage classification to mitigate the impact of daily reporting trends.

In southern Connecticut, Alpha and Iota collectively rose to above 50% prevalence by March 2021, but the relative prevalence of these variants differed across the region (Fig. 1c). Due to the close proximity of New Haven and Fairfield counties to New York City and the large volume of travelers between New York City and southern Connecticut, we hypothesized that the frequency

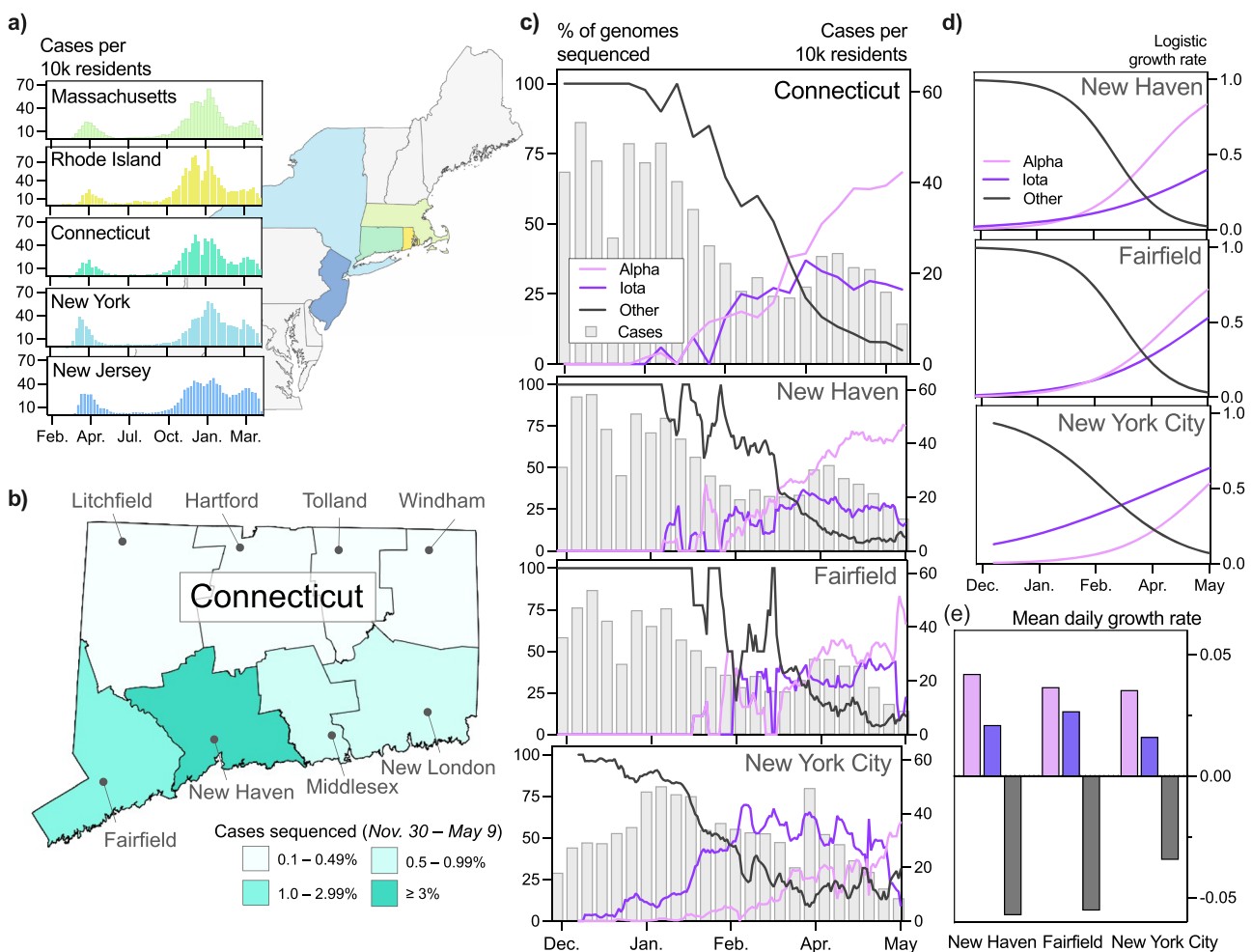

**Fig. 1 Alpha and Iota dominated the circulating SARS-CoV-2 populations in Connecticut and New York City in early 2021. a** Trends in COVID-19 incidence were consistent across northeastern states throughout the pandemic. (map) Connecticut (teal) is bordered by New York, Rhode Island, and Massachusetts. New York City is less than 50 miles from Fairfield County. Weekly COVID-19 incidence was tabulated according to the Johns Hopkins COVID-19 portal (https://github.com/CSSEGISandData/COVID-19). Shapefile source: United States Census Bureau. **b** New Haven County led the state in the percentage of COVID-19 cases sequenced between November 30, 2020 and May 9, 2021 (3.33%). During this period, 0.51% of COVID-19 cases in New York City were sequenced. Genomes that were collected through targeted variant screening (e.g., *spike*-gene target failure) were excluded from this analysis. Shapefile source: the Connecticut Department of Energy & Environmental Protection (DEEP) Geographic Information Systems Open Data Website. **c** Together, Alpha and Iota variants displaced nearly all other SARS-CoV-2 lineages in New Haven County ($n = 2086$), Fairfield County ($n = 612$), and New York City ($n = 4528$). The lineages of sequenced viruses were assigned using pangolin v.2.4.2. The lineages B.1.526, B.1.526.1, and B.1.526.2 were assigned to the general lineage category 'Iota'. We calculated a 7-day rolling average for the proportion of Alpha, Iota, and 'other' SARS-CoV-2 lineages sequenced in our dataset. **d** Daily variant incidence estimated by fitting a logistic growth model to the daily sequenced variant frequencies shown in **c**. This analysis was completed using Rv.4.0.1. Line colors correspond to the legend in **c**. **e** Daily growth rates of variants estimated using the logistic growth model shown in **d**. Bar colors correspond to the legend in **c**.

patterns in Connecticut would reflect those observed in New York City. We therefore modeled the logistic growth of each variant across locations (Fig. 1d). While the growth rates of Alpha and Iota were comparable and consistently higher than all other lineages (Fig. 1e), we observed heterogeneity in the relative growth of these variants. The estimated logistic growth rate of Alpha was twice that of Iota in New Haven County (Alpha = 0.042, Iota = 0.021) and New York City (Alpha = 0.035, Iota = 0.016). The rate of Alpha growth was 1.37 times that of Iota in Fairfield County (Alpha = 0.037, Iota = 0.028). These findings suggest that Alpha and Iota had a fitness advantage over their predecessors, and, once established, Alpha may have spread more quickly than Iota. This pattern was particularly noticeable in New York City, where Iota emerged first but increased in frequency more slowly than Alpha (Fig. 1c).

**Evidence that Alpha is more transmissible than Iota.** The relative changes in frequency and growth rates reflected by our sequencing data indicated that the growth rates of Alpha and Iota outpaced those of other co-circulating SARS-CoV-2 lineages (Fig. 1). They also provided some evidence that the prevalence of Alpha increased at a faster rate than that of Iota in three different populations. However, these observations did not account for COVID-19 incidence in each population. Over the duration of our study period, the weekly number of reported COVID-19 cases in Connecticut declined, peaking at 53 cases per 10,000 residents and falling to 9 cases per 10,000 residents with fluctuations in between. To more accurately measure the relative transmissibility of Alpha and Iota, we combined the frequency estimates from our genomic data with daily estimated COVID-19 infections and estimated the effective reproduction numbers ($R_t$),

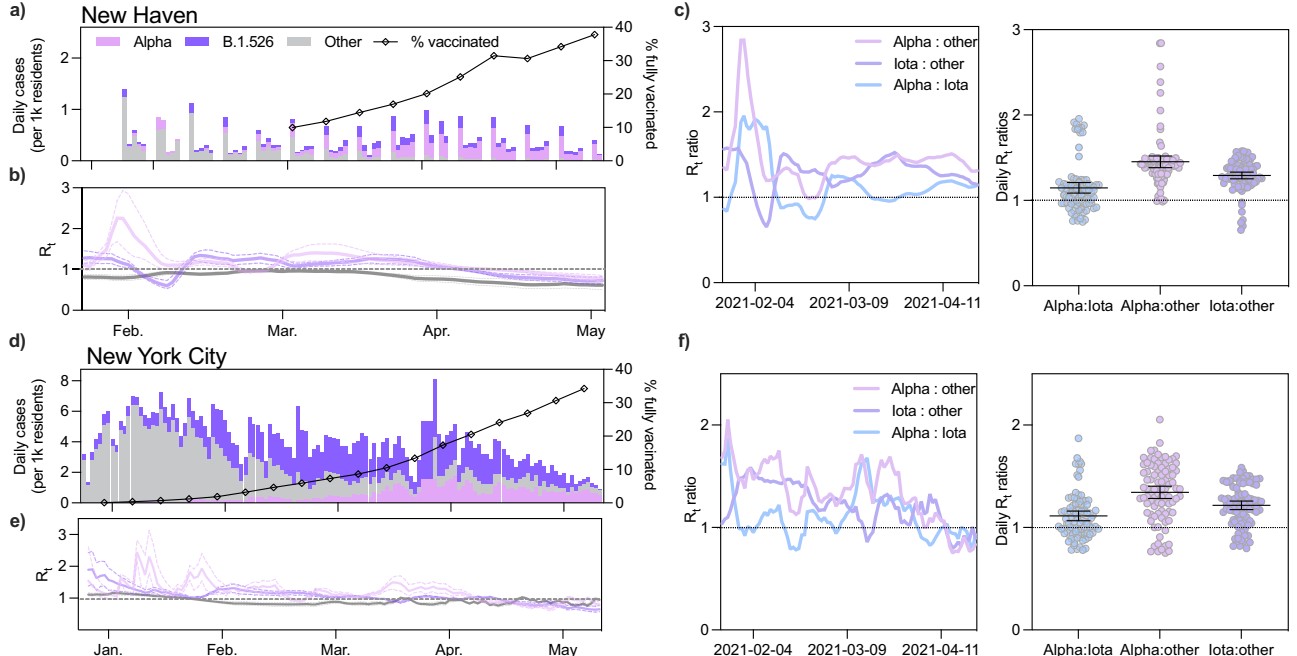

**Fig. 2 Alpha and Iota had a larger effective reproduction number ($R_t$) than other circulating lineages in the first half of 2021. a, d** Daily incidence and full vaccination rates (2 weeks post last dose) of Alpha, Iota, and other circulating lineages in New Haven County (**a**) and New York City (**d**). Estimated daily infections were assigned to one of three lineage categories ('Alpha', 'Iota', and 'other') according to the 7-day rolling average of variant frequency among sequenced cases. 'Iota' includes the sublineages B.1.526, B.1.526.1, and B.1.526.2. **b, e** Time-varying effective reproduction numbers ($R_t$) were calculated using the R package EpiEstim. An $R_t$ value above 1 indicates that an infected individual will, on average, infect more than 1 additional person. We assumed a serial interval of mean 5.2 days and standard deviation of 4 days for all lineages. 0.025 and 0.975 quantiles are shown as dotted lines. **c, f** Ratios of estimated $R_t$ between January 24 and April 24, 2021. The mean and 95% CI for the scatter plots are shown in black.

which quantifies the average number of secondary cases from a primary infection, for each variant (Fig. 2).

$R_t$ ratios of co-circulating variants reflect intrinsic differences in transmissibility. Variant-specific factors such as replication rate and immune escape interact with population-level factors including human behavior and levels of immunity to influence $R_t$ estimates. By taking the ratio of $R_t$ estimates of co-circulating variants in a given population, we can control for population effects. Therefore, using $R_t$ ratios calculated for New Haven County, Connecticut, we estimated that both Alpha and Iota were up to ~50% more transmissible than other circulating lineages (Fig. 2c). We obtained consistent albeit noisier results in New York City, providing further evidence that Alpha was more transmissible than Iota (Fig. 2f).

We estimated $R_t$ for Alpha and Iota by extrapolating the variant frequencies among sequenced cases to the total number of estimated infections in New Haven County (Fig. 2a). To estimate the number of infections in this population, we used the R package covidestim[21]. We selected New Haven County because we sequenced a higher percentage of cases compared to other counties in Connecticut (Fig. 1b), providing us with better estimates. We assumed that the 7-day rolling average of Alpha and Iota in our dataset was representative of the true prevalence of these variants in the population because these datasets were compiled using genomes collected from the same sources. Therefore, we assumed that any biases introduced through subsampling would be systematic across all lineages. However, we also calculated a Jeffreys interval for daily variant frequencies and used the 0.025 and 0.975 quantiles to compute $R_t$ and improve the robustness of our analysis (Supplementary Fig. 2).

In New Haven County, our $R_t$ estimates for Alpha and Iota followed similar decreasing trajectories as COVID-19 vaccination rates increased, though they consistently had a higher $R_t$ than

other circulating lineages (Fig. 2b). The $R_t$ for both Alpha and Iota were above 1 between February and the end of April, when fully vaccinated rates reached ~25%. An $R_t$ value above 1 indicates that on average an infected individual infects more than one additional person. The $R_t$ estimates for 'other' lineages fell below 1 in early January. Notably, the $R_t$ for Iota decreased to below 1 about one week earlier than that for Alpha (Fig. 2b). To directly compare the transmissibility of Alpha and Iota, we calculated the ratio of $R_t$ for each lineage over time (Fig. 2c). Once our estimates stabilized around the middle of February, the $R_t$ of Alpha and Iota were consistently higher than that of other lineages (Alpha range: 0.986–1.51, Iota range: 1.122–1.525) (Fig. 2c, f). The ratio of $R_t$ estimates calculated using the lower and upper quantiles of our Jeffreys intervals also exhibited this pattern (Supplementary Fig. 2). We observed a similar relationship in New York City, though with larger fluctuations (Alpha range: 0.75–1.71, Iota range: 0.80–1.47). The consistency of these findings suggests that both variants were more transmissible than other circulating lineages even when Iota emerged before Alpha. To assess the impact of reporting delays on these estimates, we compared daily $R_t$ estimates in New Haven County assuming 0 to 7 days of reporting lags and found that our estimates for Alpha remained stable, while those for Iota fluctuated slightly across our study period (Supplementary Fig. 3).

**Association of Alpha introductions with larger phylogenetic clusters than Iota introductions.** We next considered the possibility that the apparent increased transmissibility of Alpha compared to Iota was due to the number and timing of the introductions of each variant into Connecticut. More frequent introductions of Alpha could artificially inflate our $R_t$ estimates (Fig. 2). To assess this possibility, we used a Bayesian phylogeographic method to quantify the number, timing, and source of

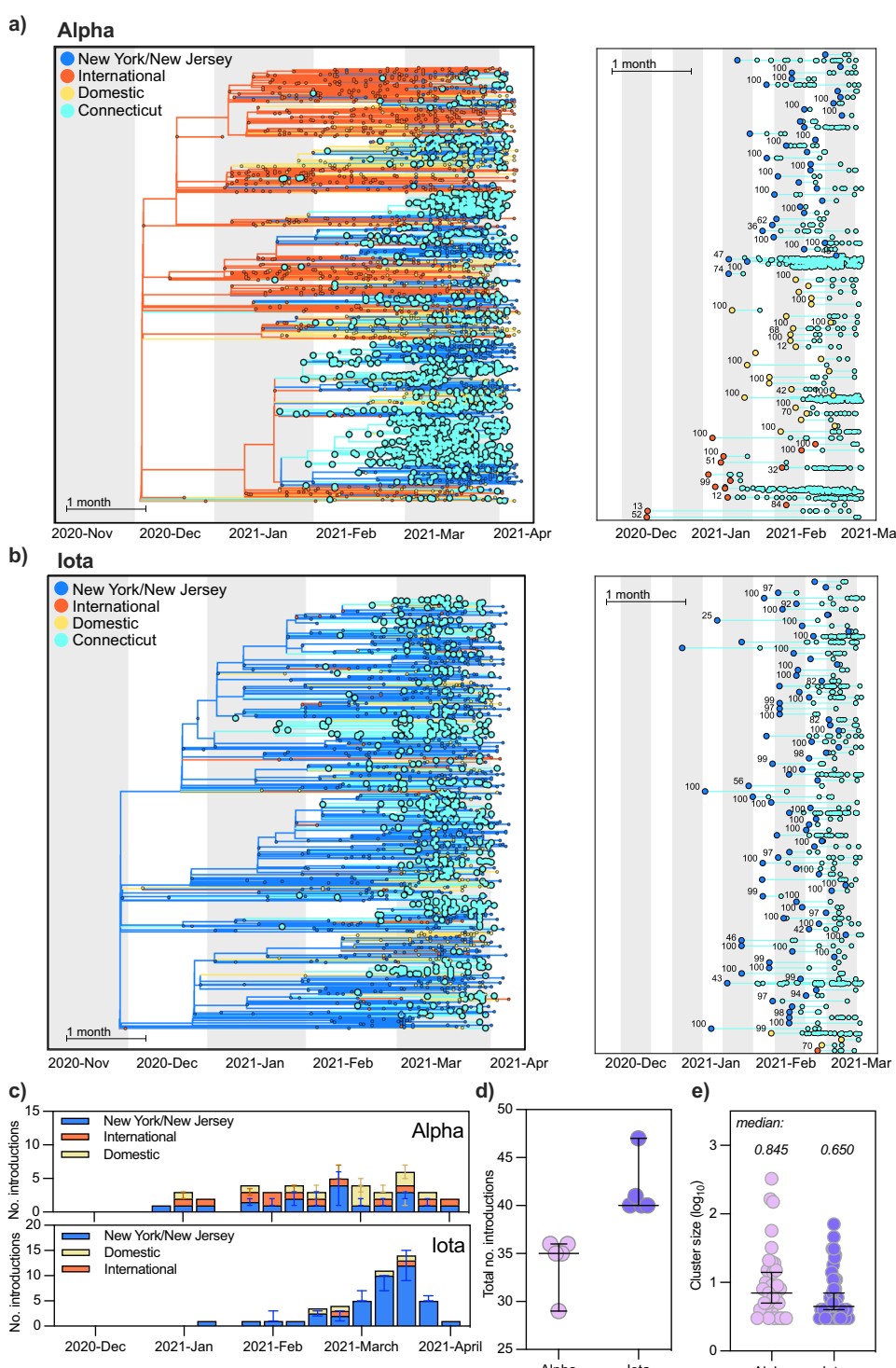

**Fig. 3 Alpha was introduced into Connecticut at a similar frequency as Iota but was associated with larger cluster sizes. a, b** Discrete phylogeography of Alpha (**a**) and Iota (**b**). Tips and nodes were assigned one of four possible locations: Connecticut, New York/New Jersey, domestic, and international. The phylogeographic analysis was performed in BEAST[23] using a time-resolved tree as the fixed topology[24]. Bootstrap values for each clade are shown at each ancestral node (right) and were obtained by constructing individual maximum likelihood trees with 1000 ultrafast bootstraps in IQTree[26]. Clades without a support value were part of polytomies. **c** We summed the number of sustained introductions for each variant by week. We defined sustained introductions as Connecticut-only clades containing at least 3 tips related by a non-Connecticut ancestor with at least 0.7 posterior probability for the inferred location. Bar colors indicate the source of introduction. Error bars show the range of the five replicates per variant. **d** There were more sustained introductions of Iota than Alpha into Connecticut. The horizontal bars show the median and 95% CI over five replicates per variant. **e** The size of Alpha clades in Connecticut was on average larger than Iota clades in Connecticut. We calculated the log₁₀ size of Connecticut clades shown in **a, b**. The horizontal lines denote the median and 95% CI log cluster size.

observed introductions of both variants into Connecticut (Fig. 3). We found that Alpha was not introduced more often into the state than Iota. Rather, the clusters resulting from each introduction were on average larger than those produced by Iota introductions. These observations are in agreement with our $R_t$ estimates and further support the likelihood that Alpha spread more rapidly than Iota.

To begin our phylogeographic analysis, we combined our SARS-CoV-2 genomic data with randomly sampled publicly available Alpha and Iota genomes (gisaid.org) from outside of Connecticut, New York, and New Jersey, normalizing by reported deaths per location (see Methods). We did this independently 5 times for each variant to account for any potential biases from the subsampling process (~17,000 and ~13,000 genomes for each Alpha and Iota subsample, respectively; Supplementary Table 2), and constructed the 10 corresponding time-resolved trees with TreeTime[22]. We next performed discrete phylogeographic reconstruction over the 10 time-resolved trees in BEAST[23,24]. We inferred the ancestral geographic states according to four discrete geographic categories: Connecticut, New York/New Jersey, domestic, and international. We chose to combine New York and New Jersey into one region because of the large volume of commuters and visitors who travel from northern New Jersey to New York City.

Due to the notably different geographic distribution of these two variants (Fig. 3a, b), we expected the source of introductions for each to also differ. Because Iota was first identified in New York and the majority of genomes from this variant family were sequenced in New York, we hypothesized New York would be the main source of Iota introductions into Connecticut. Alpha spread widely in the United Kingdom and Europe, and the first case of Alpha in Connecticut was associated with international travel[25], indicating that early introduction would likely come from international sources. We anticipated that international sources would drive the initial introductions of Alpha into Connecticut until this variant was established in the US.

We found that the sources and size of Alpha and Iota sustained introductions differed between variants and throughout the study period (Fig. 3c–e). We defined a sustained introduction as a transition from a location outside of Connecticut into Connecticut in which (**1**) the resulting clade contained at least 3 tips and (**2**) the posterior probability of the ancestral, outside-Connecticut node was at least 0.7. As we expected, New York/New Jersey was a main source of introductions of Iota into Connecticut, accounting for all but one of the 40 independent introductions (Fig. 3c–e). The sources of introductions of Alpha were heterogeneous, including international and domestic sources throughout the study period (Fig. 3c–e). The relatively limited role of New York/New Jersey in the spread of Alpha into Connecticut may be due to the lower prevalence of this variant in New York City for the majority of our study period (Fig. 1c). Our phylogeographic analysis also revealed that, although Alpha was introduced slightly less frequently than Iota (Fig. 3d), Alpha introductions led to larger clusters (Fig. 3e). These patterns were consistent across all five phylodynamic replicates and with our estimates of $R_t$ suggesting that Alpha had a fitness advantage over Iota (Supplementary Fig. 4, Fig. 2c).

## Discussion

In this study, we quantified the relative fitness of two SARS-CoV-2 variants of public health concern, Alpha and Iota, that co-circulated in New Haven, Connecticut, at the beginning of 2021. Our $R_t$ estimates indicate that both Alpha and Iota likely had a fitness advantage over other lineages, while our phylogenetic data suggest Alpha was more transmissible than Iota (Figs. 2, 3).

Importantly, we were able to control for population-level factors that may influence variant transmissibility by studying contemporaneous variant dynamics in a well-defined population. These conclusions were consistent with those of our phylogeographic analysis (Fig. 3), the typical albeit more computationally-intensive method for evaluating the dynamics of virus transmission and spread. Our analytical approach, which measures changes in frequencies and estimates the effective reproduction number for individual variants, is more informative than the current practice of tracking variant prevalence because it accounts for both the change in lineage frequencies and the number of incident cases. Moreover, it can be used to monitor the epidemiological dynamics of variants that have since emerged[27].

To demonstrate this, we also analyzed the dynamics of Alpha and Iota in New York City, where both variants also co-circulated. Our results were consistent with those from Connecticut (Fig. 2) but had some discrepancies with previous reports of relative growth rates in New York City. Specifically, West et al. reported that Iota with the *spike* E484K substitution was growing at a faster rate than Alpha in New York City between December 2020 and March 2021[12]. We also observed this rapid rise in Iota prevalence during that time period (Fig. 1d); however, we found that the growth rate of Iota slowed shortly thereafter (Fig. 1d), and Alpha became the dominant circulating lineage in April (Fig. 1c). Moreover, we estimated that the effective reproduction number of Alpha was equal to or greater than that of Iota by March (Fig. 2e), an early indicator of the eventual rise in Alpha prevalence a few months later. This second finding is particularly crucial because it illustrates that while variant frequencies at specific time points may not be indicative of variant fitness, the changes in these frequencies can reveal relative variant transmission dynamics when combined with daily incidence data.

The epidemiological findings from our case study also have broader public health implications as new SARS-CoV-2 variants continue to emerge worldwide. The sources of introductions of novel variants reflect their global distribution (Fig. 3a–c), which will likely change over time. This heterogeneity poses a serious obstacle to control and prevention efforts because it limits the efficacy of policies that target specific points of entry. For variants that are prevalent on multiple continents like Alpha and, more recently, Delta and Omicron, testing, contact tracing, and vaccination campaigns within communities will likely prove more efficient in limiting their spread than targeting a specific subset of travelers. Once local transmission of a new variant has been established, assessing the public health threat is both challenging and necessarily retrospective. However, a robust genomic surveillance infrastructure coupled with the application of a framework like ours would enable the routine monitoring of variant epidemiology. The phylodynamic methods we applied in this study can be run on a desktop computer in a few hours, making the collation of representative datasets the rate limiting step. Expanding genomic surveillance efforts would remove this barrier and promote a rapid and efficient response to outbreaks caused by new variants.

There were some limitations to the epidemiological findings we have presented. First, we were not able to directly measure the secondary attack rates of individuals infected with Alpha or one of the Iota sublineages. Collecting this information requires extensive contact tracing and sequencing of all secondary infections that are not available in Connecticut. Instead, we assumed that biases introduced by the method we employed in this study would be systematic across SARS-CoV-2 lineages so that estimates of the relative transmissibility of Alpha and Iota would be unaffected. Similarly, although we allowed the generation interval times to vary in our $R_t$ estimations, we did not explicitly account for potential differences in generation interval distributions

between the two variants due to the limited data availability on this point. However, we believe our estimates to be accurate because we found that Alpha was ~50% more transmissible than non-Iota lineages, and this is consistent with previously published findings by Washington et al.[11]. Second, we did not evaluate the individual dynamics of the previously designated Iota sublineages: B.1.526, B.1.526.1, and B.1.526.2. While there may be variation in the $R_t$ of the individual sublineages, we elected to capture the reproduction number of Iota as per its designation by the WHO. Third, we used a small subset of publicly available SARS-CoV-2 genomes for our phylodynamic analyses to make them computationally tractable. Incorporating a small proportion of available data into our analyses may have introduced biases, but by demonstrating the reproducibility of our findings with independent replicates (Supplementary Fig. 3), we substantially mitigated this issue. The use of publicly available data also introduced the potential for uncontrolled geographic sampling biases in our phylogeographic analysis. However, because the majority of available Connecticut genomes were generated at Yale University or Jackson Laboratory, we assumed that biases would be systematic across variants. Fourth, performing phylogeographic inference using a fixed topology may overestimate the number of introduction events because this method does not resolve polytomies. Finally, the scope of our study was limited to Connecticut and, in some cases, New York City, which may impinge upon the generalizability of our findings. However, our objective was to directly compare the fitness of Alpha and Iota, and Connecticut is one of few locations with a robust genomic surveillance infrastructure where these variants emerged concurrently.

Here, we use genomic data to estimate the effective reproduction number of two co-circulating SARS-CoV-2 variants as a measure of relative transmission and fitness. By focusing on Connecticut, this study directly compares the fitness of Alpha and Iota in a setting where they emerged concurrently. Our analysis of Alpha and Iota dynamics in New York City not only corroborates our findings in Connecticut, but also provides insight into the SARS-CoV-2 populations circulating through the Connecticut-New York corridor, which is an international travel hub. Moreover, our findings highlight that many factors influence a variant's success including the timing of introduction, the existing virus population, host immunity, and advantageous amino acid substitutions. As new SARS-CoV-2 variants emerge, it will be critical to assess the magnitude of the role that each of these elements play in precipitating local outbreaks so that appropriate, effective, and immediate steps may be taken to control further SARS-CoV-2 transmission.

## Methods

### Ethics

*Yale university*. The Institutional Review Board from the Yale University Human Research Protection Program determined that the RT-qPCR testing and sequencing of de-identified remnant COVID-19 clinical samples obtained from clinical partners conducted in this study is not research involving human subjects (IRB Protocol ID: 2000028599).

*Jackson laboratory*. The Institutional Review Board of The Jackson Laboratory determined that use of de-identified residual COVID-19 clinical samples obtained from the Clinical Genomics Laboratory for RT-qPCR testing and sequencing for this study is not research involving human subjects (IRB Determination: 2020-NHSR-021).

*New York State Department of Health, Wadsworth Center*. Residual portions of respiratory specimens from individuals who tested positive for SARS-CoV-2 by RT-PCR were obtained from the Wadsworth Center and partnering clinical laboratories. This work was approved by the New York State Department of Health Institutional Review Board, under study numbers 02-054 and 07-022.

### Reported COVID-19 case data

We used daily reported cases compiled by the Johns Hopkins COVID-19 portal (https://github.com/CSSEGISandData/COVID-19).

We summed the number of incident cases by week by state for Massachusetts, New York, Rhode Island, New Jersey, and Connecticut, and we aggregated incident cases by week by county for New Haven, Fairfield, and Westchester. We visualized these data using Prism v.9.0.2 (plots) and Rv.1.2 (maps). For the latter, we obtained the shapefiles from the United States Census Bureau (east coast) and the Connecticut Department of Energy & Environmental Protection (DEEP) Geographic Information Systems Open Data Website (Connecticut).

### SARS-CoV-2 sequencing and consensus generation

*Yale university*. We received clinical samples from confirmed SARS-CoV-2 positive individuals from routine testing provided by Yale New Haven Hospital, Yale Pathology Laboratory, "Yale Campus Study", Connecticut Department of Public Health, and Murphy Medical Associates. These samples were sent as either nasal swabs in viral transport media, raw saliva, or extracted and purified RNA. For the former two, we extracted RNA from 300 μl of the original sample using the MagMAX viral/pathogen nucleic acid isolation kit, eluting in 75 μl of elution buffer. We tested the extracted nucleic acid using our 'variant of concern' RT-qPCR assay to determine the SARS-CoV-2 viral RNA load[28]. Samples with cycle thresholds <35 were prepared for sequencing using the Illumina COVIDSeq Test RUO version to synthesize cDNA, and generate and tagment amplicons. Amplicons were pooled and cleaned before quantification with Qubit High Sensitivity dsDNA kit. The resulting libraries were sequenced using a 2 × 100 or 2 × 150 approach on an Illumina NovaSeq at the Yale Center for Genomic Analysis. Each sample was given at least 1 million reads. Samples were typically processed in sets of 94 with negative controls incorporated during the RNA extraction, cDNA synthesis, and amplicon generation steps.

Using BWA-MEM v.0.7.15[29], we aligned reads to the Wuhan-Hu-1 reference genomes (GenBank MN908937.3). With iVar v1.2.1[30] and SAMtools[31], we trimmed sequencing adapters, masked primer sequences, and called bases by simple majority (>50% frequency) at each site to generate consensus genomes. An ambiguous 'N' was used when fewer than 10 reads were present at a site. In all cases, negative controls were analyzed and confirmed to consist of at least 95% Ns. We used pangolin v.2.4.2[18] to assign lineages[20]. Consensus genomes were uploaded to GISAID.

*Jackson laboratory*. Clinical samples were received in The Jackson Laboratory Clinical Genomics Laboratory (CGL) as part of a statewide COVID-19 surveillance program, with the majority of samples representing asymptomatic screening of nursing home and assisted living facility residents and staff. Total nucleic acids were extracted from anterior nares swabs in viral transport media or saline (200 μl) using the MagMAX Viral RNA Isolation kit (ThermoFisher) on a KingFisher Flex purification system. Samples were tested for the presence of SARS-CoV-2 RNA using the TaqPath COVID-19 Combo Kit (ThermoFisher). Samples with cycle thresholds ≤30 for the N gene target were prepared for sequencing using the Illumina COVIDSeq Test kit. Sequencing was performed on an Illumina NovaSeq or NextSeq in the CGL. Data analysis was performed using the DRAGEN COVID Lineage App in BaseSpace Sequence Hub. Sequences with >80% of bases with non-N basecalls and ≥1500-fold median coverage were considered successful and were submitted to GISAID. Lineages were assigned using pangolin v.2.4.2[18] and the most current version of the pangoLEARN assignment algorithm.

*New York State Department of Health, Wadsworth Center*. Respiratory swabs positive for SARS-CoV-2 were sent to the Wadsworth Center from collaborating clinical laboratories across New York State as part of an enhanced genomic surveillance program initiated by the New York State Department of Health in December 2020. Specimens were required to have a real-time cycle threshold value less than 30. Nucleic acid extraction was performed on a Roche MagNAPure 96 (Roche, Indianapolis, IN) and RNA was processed for whole genome sequencing with a modified ARTIC3 protocol (http://artic.network/ncov-2019) in the Applied Genomics Technology Core at the Wadsworth Center, by adding additional ARTIC3 primers when poor amplification efficiency was observed[10]. Lineage was determined by GISAID using pangolin software[18], updated June 7, 2021. Daily relative frequency of variants within New York City was determined based on sample collection date and patient residence within Bronx, Kings, New York, Queens, or Richmond counties. Any specimens that were sequenced as a result of pre-screening for specific mutations or clinical/epidemiological criteria were removed from the analysis. Consensus genomes were uploaded to GISAID.

### Percent of COVID-19 cases sequenced

To calculate the percent of cases sequenced in each county, we tabulated the number of genomes collected from the state with available county-level data. Though this level of geographic resolution was only available for genomes sequenced by our laboratory and the Jackson Laboratory for Genomic Medicine, these two sources have generated the vast majority of genomes for the state of Connecticut. For New York City, NY, we used genomes generated by the Wadsworth Center. Using the case data described above, we summed the number of cases reported by each county between November 30, 2020 and May 9, 2021, and divided the total number of genomes generated for each county within the same timeframe by that sum.

**Frequency of SARS-CoV-2 variants among sequenced cases**. To assess the frequency of circulating lineages, we selected genomes that were sequenced through a non-biased sampling approach. Specifically, we excluded genomes that were screened and sequenced through a targeted S-gene target failure surveillance system. As with the dataset we used to measure the percent of cases sequenced by county, these genomes were generated by our laboratory, Jackson Laboratory, and the Wadsworth Center. We organized these genomes into three categories using Pangolin v.2.4.2[18]: Alpha, Iota*, and 'other'. We then tabulated the number of genomes in each category by week and calculated the percent of the total number of genomes for that week.

**Distribution of SARS-CoV-2 variants among cases**. We obtained estimates of the distribution of cases attributed to each lineage category by multiplying the frequency of that category by the number of cases reported in the same week. In doing so, we assumed that the sequencing frequencies described above were representative of the virus population circulating in New Haven and Fairfield counties, and New York City (all counties). We also assumed that the number of reported cases for each county was representative of the true number of infections in that region.

To account for any uncertainty in our assumption that the sampling frequencies were representative of cases per county, we began by calculating p, a 7-day rolling average for the proportion of sequenced cases for each lineage category. This produced daily proportion estimates. To further account for any uncertainty, for each p, we calculated a Jeffreys interval, which is a Bayesian, equal-tailed interval of the form[32]:

$$quantile = \beta(x + 0.5, n - x + 0.5) \qquad (1)$$

where β represents the beta distribution, x represents the 7-day rolling average of sequences of a specific lineage, and n represents the 7-day rolling average of sequences for all lineages. Our measure of interest, p, is calculated by x/n. The Jeffreys intervals were calculated using the package "DescTools" in R v4.0.1.

**Logistic regression**. We computed logistic growth models for each lineage category in each county using the frequency estimates described above. Specifically, we fitted a generalized linear model using a binomial distribution to our frequency estimates in R v4.0.1.

**Effective reproduction number**. Using p, and the 0.025 and 0.975 quantiles from the Jeffreys interval, we multiplied these values by the number of estimated infections per day. We estimated infections using the R package covidestim[21]. These three potential infection counts were used to calculate the reproduction number ($R_t$), the mean number of secondary cases generated by a typical primary case at time t in a population. Further, for the $R_t$ distribution calculated from p, we also computed the 0.025 and 0.975 quantiles (Supplementary Fig. 2).

Because there is no consensus in the literature as to the precise serial interval for each variant, we used an uncertain serial interval with mean of 5.2 days and standard deviation of 4 days[33–35]. Through the uncertain serial interval, multiple distributions were explored where the mean was allowed to vary from 2.2 to 8.2 days, and the standard deviation varied from 2.5 to 5.5 days. From each of these $R_t$ distributions, we selected the median $R_t$ to represent a given lineage's instantaneous effective reproductive number per day. All of the $R_t$ estimates were calculated using the "EpiEstim" package in R v4.0.1.

To evaluate the impact of reporting delays on $R_t$ estimates, we used the R package 'EpiNow2' to estimate $R_t$ assuming 0 to 7 days of reporting delays (Supplementary Fig. 3).

**COVID-19 vaccination rates**. We obtained vaccination data for New York City from data.cdc.gov and for Connecticut from data.ct.gov (COVID-19 Vaccinations by Town and Age Group).

**SARS-CoV-2 genome selection for phylogenetic analysis**. We downsampled both Alpha and Iota datasets using COVID-19 death counts. We elected to normalize genome counts to the number of deaths because deaths are less likely to be under-reported than cases[36]. We obtained daily death counts for all countries and US states from the Johns Hopkins COVID-19 portal (https://github.com/CSSEGISandData/COVID-19). We summed the cumulative number of deaths for each state or country between October 1, 2020 and April 23, 2021 because we assumed that deaths could not be attributed to either variant before October 1. Because country-level death data were not reported for countries within the UK, we calculated the total number of genomes to sample from the UK with the method described above, and then calculated the distribution of genomes by UK country based on population size. To calculate the number of genomes we would include in our final datasets from each region, we first calculated the ratio of variant genomes to deaths. In doing so, we assumed that the number of variant genomes sequenced by each country or state was proportional to the total number of genomes sequenced in those places.

For non-US sampling, if the number of variant genomes sequenced comprised less than 1% of cumulative deaths, we included all of the genomes from that location. Otherwise, we selected the number of genomes that corresponded to 1%

of reported cumulative deaths. We used a similar approach for US states except we set the minimum threshold to 0.1% of cumulative deaths. In all cases, if a country or state had less than 20 genomes available, we included all of them. For the Iota lineages, we calculated the proportion of each lineage out of the total number of Iota genomes sequenced in each country or state and selected genomes according to this proportion.

We did not downsample Connecticut, New York, or New Jersey for either variant dataset in the first stage of downsampling. Once functional duplicates were removed from these locations, we included 1,408 genomes sequenced by Yale (Connecticut), 497 sequenced by Jackson Laboratory (Connecticut), and 803 sequenced by the Wadsworth Center (New York City) collected between December 1, 2020 and April 23, 2021. We obtained all other genomes for this analysis from GISAID (gisaid.org). We also applied a modified sampling scheme for Alpha genomes from Australia, New Zealand, Sint Maarten, Bonaire, Vietnam, or Singapore because these locations reported a negligible number of deaths. For this reason, it was impossible to downsample based on the number of reported deaths. We therefore randomly selected 1% of available genomes from those locations instead. To select the genomes to incorporate into our dataset from the downsampled locations, we randomly selected a weekly set of genomes equal to 1% of deaths per week. Using this workflow, we generated five datasets for each variant to serve as independent replicates for the remainder of our analysis. In all cases, we excluded genomes containing more than 30% Ns from our selection. Due to the broader global distribution of Alpha, the datasets for this variant were necessarily larger than those for Iota.

At that stage, the datasets were still too large to be computationally tractable. We next scaled each dataset by a factor of 0.1 by randomly selecting 10% of genomes by country or state (US only). We did not scale genomes from Connecticut so that the final datasets were not precisely one tenth the size of the original (Supplementary Table 2).

### SARS-CoV-2 phylogenetic analysis

*Sequence alignment and refinement.* Having compiled our ten datasets, we aligned the genomes using MAFFT[37]. We then removed gaps and masked problematic sites[38]. We then removed functional duplicates from each dataset to reduce phylogenetic redundancy. We defined a functional duplicate as genomes that shared identical sequences, week of collection, and geographic region. For genomes collected in Connecticut and New York, we defined the geographic region as the county. For genomes collected elsewhere in the US, we defined it as 'state'. For genomes collected internationally, we defined the geographic region as 'country'.

*Maximum likelihood construction.* To identify and remove problematic genomes from our datasets, we performed a preliminary phylogenetic analysis in IQTree[26]. Each tree was rooted using a P.1 genome (hCoV-19/Brazi/AM-FIOCRUZ-20842882CA/2020). We performed a root-to-tip analysis in TempEST[39] and removed outliers with residuals > |0.0015 |. We constructed a maximum likelihood tree with each dataset ($n = 10$) using a GTR+G substitution model with 1000 ultrafast bootstraps again with IQTree.

*Time-resolved construction.* To avoid computational bottlenecks in our phylogeographic reconstruction, we did not use a Bayesian method to infer the temporal resolution of our maximum likelihood tree. We have previously shown that temporal estimates inferred using TreeTime agree with those inferred from BEAST for Alpha[10]. We used the bootstrapped trees and associated alignments to construct corresponding time-resolved phylogenetic trees with TreeTime v.0.8.0[22]. This method is implemented in an augur pipeline[40].

*Discrete phylogeographic analysis.* We elected to use a Bayesian approach to infer geographic ancestral states because we aimed to identify 'sustained introductions' of each variant into Connecticut. We defined a sustained introduction as a transition from a non-Connecticut state to a Connecticut state with at least 0.7 posterior probability for the inferred location with clade containing at least 3 tips.

We performed a discrete phylogeographic analysis with the time-resolved trees as the fixed topology using BEAST[23,24]. Specifically, we assigned a location to each of the tree tips from four categories: 'Connecticut', 'New York/New Jersey', 'domestic', and 'international'. We used an asymmetric substitution model and a strict clock to model location. We ran each tree for 1 million chains and used Tracer v.1.7.1 to confirm that all parameters had achieved ESS values of at least 200.

We identified Connecticut-only clades and their source of introduction using the "exploded tree" script implemented with baltic 0.1.6 (https://github.com/evogytis/baltic). We restricted our subsequent analysis to clades that represented sustained introductions. We aggregated the number of sustained introductions by week and source, and visualized the results using Prism v.9.0.2. We merged the bootstrap values from our original trees with the topology of our geographically-resolved trees using baltic.

*Statistics and reproducibility.* All statistical analyses were performed using R v.4.0.1. The discrete phylogeographic analysis was completed in 5 replicates per variant using downsampled subsets of publicly available SARS-CoV-2 genomic data.

**Reporting summary**. Further information on research design is available in the Nature Research Reporting Summary linked to this article.

## Data availability

All of the genomic data used for the analyses in this manuscript are available on GISAID (gisaid.org). We gratefully acknowledge all of the laboratories that obtained the clinical specimen and generated the SARS-CoV-2 genomes used in our analyses. All files associated with our phylogenetic analysis, including full acknowledgment of the laboratories whose genomes we used, may be found in our Figshare repository (DOI: 10.6084/m9.figshare.c.5928089).

## Code availability

Generalized code used to generate effective reproduction number estimates may be found in our Github repository (https://github.com/grubaughlab/paper_2021_B117vsB1526; https://doi.org/10.5281/zenodo.6403875).

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

## Acknowledgements

We thank the frontline and essential workers for their service during the pandemic, the groups that continuously make their data available to the public, K. Gangavarapu for technical advice, C. O'Connor for map creation, and our friends and family - particularly V. Parsons, P. Jack, S. Currie, and S. Taylor—for their support. This work was funded by CTSA Grant Number TL1 TR001864 (M.E.P. and T.A.), Fast Grant from Emergent Ventures at the Mercatus Center at George Mason University (N.D.G.) and CDC Contract # 75D30120C09570 (N.D.G.). Initial funding for sequencing at the Wadsworth Center was provided by the New York Community Trust.

## Author contributions

M.E.P., J.E.R., B.P.T., W.P.H., L.M.G., V.E.P., and N.D.G. conceptualized this project. K.K., G.O., N.R., R.E., S.M., C.N., E.L., A.M., R.D., J.R., L.N., M.S.W., M.L.A., J.W., C.L., P.H., M.L.L., D.R.P., and M.D.A. contributed to sample acquisition and diagnostic test. M.E.P., M.I.B., I.M.O., A.R., E.L-N., K.K., G.O., N.R., A.E.W., C.C.K., T.A., A.F.B., R.E., I.R.T., C.C., J.P.K., M.S., J.P., E.S., and J.R.F. performed sequencing and data processing. M.E.P., J.E.R., A.R., L.M.G., V.E.P., and N.D.G. analyzed and interpreted the data. S.M., K.B., J.R.F., C.B.F.V., L.M.G., V.E.P., K.S.G., M.D.A., and N.D.G. supervised this project. The original draft of the manuscript was written by M.E.P. and N.D.G. J.E.R., A.R., C.B.F.V., and M.D.A. reviewed and edited the manuscript.

## Competing interests

N.D.G. is a paid consultant for Tempus Labs to develop infectious disease diagnostic assays. K.S.G. receives research support from Thermo Fisher for the development of

assays for the detection and characterization of viruses. All other authors declare no competing interests.
