## [Peer Review File · Communications Biology]

REVIEWERS' COMMENTS:

Reviewer #1 (Remarks to the Author):

This is a much improved manuscript by Petrone and colleagues as they have now re-focused the manuscript on comparing Alpha and Iota rather than the original "real-time" framework previously presented. The authors have been very diligent in addressing prior concerns and have made robust defenses against prior comments. As such I can only commend the authors for addressing these concerns and modifying the manuscript accordingly. I think the addition of the maximum likelihood and time resolved trees used as input for BEAST and the respective XML files for each BEAST run may help other groups as well as the R scripts.

Reviewer #2 (Remarks to the Author):

I am pleased with the modifications the authors have made to their manuscript. The methods are clearer and the results more suitably contextualized.

I must disagree with the author's rebuttal that "It is therefore no longer possible to call any of the sublineages" of B.1.526. Although it is true that PANGO no longer does this, it can be done by looking at the mutations within genomes themselves and/or building a phylogenetic tree (as they have done in Figure S1). And some Iota lineages, like 484E, did decline during their study period. I realize that the authors do not wish repeat their entire study, breaking up B.1.526 into multiple sub-analyses. And I am not requesting that they do this. However, their justification for not reanalyzing the Iota data is disingenuous. As a compromise, I would strongly encourage the authors to further couch their findings, acknowledging that variation in R_t may exist within Iota and that this study was not designed to detect these differences.

Once this limitation has been acknowledged, I believe this manuscript will be suitable for publication.

Reviewer #3 (Remarks to the Author):

After reviewing the manuscript in its current form and the authors detailed response to previous reviewers' comments, I believe the authors have adequately addressed the concerns raised during the last round of peer-review process. I also want to highlight a few strengths of the paper that may have been missed by other reviewers. In my opinion, the authors strike a good balance between how epidemiologic and phylogenetic tools are being deployed. And the epidemiologic and genomic data are being utilized in a way that provide "orthogonal" evidence truly complimentary to each other. The study generated insights such as variant-specific R_t estimates to measure variant fitness over time by using sequencing data to decompose incidence time series then apply state-of-the-art epidemiologic tool to estimate R_t . It then utilizes phylogeographic analysis to estimate, for each variant, the number of independent introductions and the cluster size of each introduction. The relative fitness among variants can also be evaluated based on the cluster size of variant-specific introductions. The potential bias of "founder-effect" has also been addressed through controlling for introductions. This is important as the authors analysis suggests Alpha and Iota have similar transmission advantage, but Alpha variant may have a slight edge over Iota. This claim would have been unconvincing by epidemiologic or genomic evidence alone due to the noisy nature of the data. But since two independent lines of evidence derived from completely different methodologies point to a same conclusion makes a much stronger case. In addition, the authors made a very interesting observation that, while Iota and Alpha have similar fitness advantage, the Alpha variant raised to global domination while Iota circulation were more confined geographically. This is clearly illustrated in Figure 3 showing Alpha's introductions into Connecticut were "global" while Iota's introductions were mostly linked to New York, likely reflecting a global "founder effect" for Alpha and local (New York) "founder effect" for Iota. Overall, I believe the paper demonstrate a powerful application of "genomic epidemiology" in advancing our understanding on SARS-CoV-2 circulation dynamics at variant level, which is highly relevant in the era of Delta and Omicron

waves. I would like to recommend the manuscript for publication once the following cosmetic issues were addressed:

- Figure 1(d): I'm not really sure what's being presented in this panel. The title says Logistic growth rate, but the plot looks more like the estimated variant prevalence through fitting a logistic growth model. Please clarify.
- Figure 1(e): the slopes of the logistic growth should be the growth rates, and growth rates should have units. Please indicate unit (per day?) on the y axis.
- Figure 2(c, f): please consider overlay boxplot on top of the scatter. It's more informative to have the data's point estimate, interquartile, and range visualized than just the raw, overlapping, data points.
- Same as above but for Figure 3(e).

Reviewer #1

This is a much improved manuscript by Petrone and colleagues as they have now re-focused the manuscript on comparing Alpha and Iota rather than the original "real-time" framework previously presented. The authors have been very diligent in addressing prior concerns and have made robust defenses against prior comments. As such I can only commend the authors for addressing these concerns and modifying the manuscript accordingly. I think the addition of the maximum likelihood and time resolved trees used as input for BEAST and the respective XML files for each BEAST run may help other groups as well as the R scripts.

We thank the reviewer for their kind words and assistance on improving this manuscript.

Reviewer #2

I am pleased with the modifications the authors have made to their manuscript. The methods are clearer and the results more suitably contextualized.

We thank the reviewer for their positive feedback.

I must disagree with the author's rebuttal that "It is therefore no longer possible to call any of the sublineages" of B.1.526. Although it is true that PANGO no longer does this, it can be done by looking at the mutations within genomes themselves and/or building a phylogenetic tree (as they have done in Figure S1). And some Iota lineages, like 484E, did decline during their study period. I realize that the authors do not wish repeat their entire study, breaking up B.1.526 into multiple sub-analyses. And I am not requesting that they do this. However, their justification for not reanalyzing the Iota data is disingenuous. As a compromise, I would strongly encourage the authors to further couch their findings, acknowledging that variation in R_t may exist within Iota and that this study was not designed to detect these differences.

Once this limitation has been acknowledged, I believe this manuscript will be suitable for publication.

We appreciate the reviewer's feedback on this topic.

We have added the following qualification to the limitations section of our discussion section:

"We did not evaluate the individual dynamics of the previously designated Iota sublineages: B.1.526, B.1.526.1, and B.1.526.2. While there may be variation in the R_t of the individual sublineages, we elected to capture the reproduction number of Iota as per its designation by the WHO."

Reviewer #3

After reviewing the manuscript in its current form and the authors detailed response to previous reviewers' comments, I believe the authors have adequately addressed the concerns raised during the last round of peer-review process. I also want to highlight a few strengths of the paper that may have been missed by other reviewers. In my opinion, the authors strike a good balance between how epidemiologic and phylogenetic tools are being deployed. And the epidemiologic and genomic data are being utilized in a way that provide "orthogonal" evidence truly compliment to each other. The study generated insights such as variant-specific R_t estimates to measure variant fitness over time by using sequencing data to decompose incidence time series then apply state-of-the-art epidemiologic tool to estimate R_t . It then utilizes phylogeographic analysis to estimate, for each variant, the number of independent introductions and the cluster size of each introduction. The relative fitness among variants can also be evaluated based on the cluster size of variant-specific introductions. The potential bias of "founder-effect" has also been addressed through controlling for introductions. This is important as the authors analysis suggests Alpha and Iota have similar transmission advantage, but Alpha variant may have a slight edge over Iota. This claim would have been unconvincing by epidemiologic or genomic evidence along due to the noisy nature of the data. But since two independent lines of evidence derived from completely different methodologies point to a same conclusion makes a much stronger case. In addition, the authors made a very interesting observation that, while Iota and Alpha have similar fitness advantage, the Alpha variant raised to global domination while Iota circulation were more confined geographically. This is clearly illustrated in Figure3 showing Alpha's introductions into Connecticut were "global" while Iota' introductions were mostly linked to New York, likely reflecting a global "founder effect" for Alpha and local (New York) "founder effect" for Iota. Overall, I believe the paper demonstrate a powerful application of "genomic epidemiology" in advancing our understanding on SARS-CoV-2 circulation dynamics at variant level, which is highly relevant in the era of Delta and Omicron waves. I would like to recommend the manuscript for publication once the following cosmetic issues were addressed.

We thank the reviewer for taking the time to highlight the strengths of our study!

Figure 1(d): I'm not really sure what's being presented in this panel. The title says Logistic growth rate, but the plot looks more like the estimated variant prevalence through fitting a logistic growth model. Please clarify.

The reviewer is correct that this figure was not labeled completely correctly.

We have updated the figure caption to read:

"Daily variant incidence estimated by fitting a logistic growth model to the daily sequenced variant frequencies shown in (c)."

Figure 1(e): the slopes of the logistic growth should be the growth rates, and growth rates should have units. Please indicate unit (per day?) on the y axis.

We have corrected the figure caption for 1(e) as follows:
“Daily growth rates of variants estimated using the logistic growth model shown in (d).”

We have also updated the label on the Y-axis:

Figure 2(c, f): please consider overlay boxplot on top of the scatter. It's more informative to have the data's point estimate, interquartile, and range visualized than just the raw, overlapping, data points.

We have updated Figure 2 (c,f) accordingly:

We have updated the figure caption to reflect this change:
“The mean and 95% CI for the scatter plots are shown in black.”

Same as above but for Figure 3(e).

We have updated Figure 3(e) and its caption:

“The horizontal lines denote the median and 95% CI log cluster size.”